# Chitosan or Cyclodextrin Grafted with Oleic Acid Self-Assemble into Stabilized Polymeric Micelles with Potential of Drug Carriers

**DOI:** 10.3390/life13020446

**Published:** 2023-02-04

**Authors:** Igor D. Zlotnikov, Dmitriy A. Streltsov, Natalya G. Belogurova, Elena V. Kudryashova

**Affiliations:** Faculty of Chemistry, Lomonosov Moscow State University, Leninskie Gory, 1/3, 119991 Moscow, Russia

**Keywords:** chitosan micelle, moxifloxacin, rifampicin, enhancing drugs, overcoming pathogen resistance

## Abstract

Polymeric micelles combining the advantages of biocompatible poly- and oligosaccharides with classical micellar amphiphilic systems represent a promising class of drug carriers. In this work, micelles based on chitosan (or cyclodextrin) and oleic acid with various modification degrees were synthesized—the most optimal grafting degree is 15–30% in terms of CMC. According to NTA data, micelles have a hydrodynamic diameter of the main fraction of 60–100 nm. The inclusion of the antibacterial agents: moxifloxacin or rifampicin in micelles was studied by FTIR spectroscopy and fluorescence spectroscopy using a pyrene label (using monomer-excimer approach). When aromatic molecules are incorporated into micelles, the absorption bands of C-H bonds of the fatty tails of micelles shift towards smaller wavenumbers, indicating a stabilization of the micelles structure, and the microenvironment of the drug molecule changes according to the low frequencies shift and intensity changes in oscillation frequencies of 1450 cm^−1^ corresponding to aromatic fragment. Loading of moxifloxacin and rifampicin into micelles leads to a change in the fluorescent properties: a shift of the maximum of fluorescence emission to the long-wavelength region and an increase in the fluorescence anisotropy due to a drastic increase in the hydrodynamic volume of the fluorophore-containing rotating fragment. Using the pyrene label, the critical micelle concentrations were determined: from 4 to 30 nM depending on the polymer composition. Micellar systems enhance the effect of the antibiotic by increasing the penetration into bacterial cells and storing the drug in a protective coat. As a part of the supramolecular structure, the antibiotic remains active for more than four days, while in free form, the activity decreases after two days. In pharmacokinetic experiments, in vivo moxifloxacin in micellar systems show 1.7 times more efficiency compared to free form; moreover, two times higher maximal concentration in the blood is achieved. The advantage of polymer micellar systems in comparison with simple cyclodextrins and chitosan, which do not so significantly contribute to the antibacterial and pharmacokinetic parameters, was shown. Thus, polymeric micelles are one of the key approaches to improving the effectiveness of antibacterial drugs and solving the problems of resistant bacterial infections and multidrug resistance.

## 1. Introduction

*Micelles are class, micelles are power, micelles are cool, micelles are Dinamo soccer school*.

Bacterial infections are one of the acute problems of our time [1,2,3]. Existing approaches to combating pathogens lead to the emergence of bacterial resistance and the inability to treat dormant or latent forms [4,5,6]. One of the ways to solve the problem is to use biocompatible delivery systems for targeted drugs and molecular containers. Various systems with individual advantages are described in the literature: cyclodextrins—low carrier weight and high drug loading [7,8,9,10,11,12], polymers—significant prolongation, increased bioavailability [13,14,15,16,17,18,19], and liposomes and micelles—dissolution of hydrophobic substances [20,21,22,23,24]. In this paper, we propose a promising option for combining the properties of oligo- and polysaccharides with the advantages of the micellar systems [19,25,26] by creating combined amphiphilic macromolecules based on the chitosan (or cyclodextrins) and fatty acids. Previously, polymeric micelles developed by creating bi- and trifunctional conjugates have been proven as promising systems for the delivery of hydrophobic drugs [27,28,29,30,31].

This paper describes the experimental basis for the creation of drug delivery systems with variable parameters. The choice of oleic acid and chitosan is based on their properties and literature data on biocompatibility, high mass-loading of drug, prolonged action, and a wide range of substances that can be included in micelles. The choice of polymer systems rather than inorganic nanoparticles is due to the better bioavailability, stability, and low toxicity of the former [32]. A robust core–shell structure, kinetic stability, and the inherent ability to solubilize hydrophobic drugs are the highlights of polymeric micelles [33,34]. Cytotoxicity tests on human dermal fibroblasts demonstrated good biocompatibility of Chitosan-oleic conjugates especially (and to a lesser extent linoleic acid derivatives) [35]. For Chit-OA conjugates, it was shown that they are easily self-organizing into a micelle-like structure in an aqueous medium, have a low critical micellar concentration of 1 μg/mL, allow the dissolution of poorly soluble drugs, maintain the permeability of micelles across the gastrointestinal tract, and improve pharmacokinetic parameters (maximum concentration up to 2-fold and AUC up to 2.6-fold compared to free drug suspension [36].

The variability of the system parameters in the future is important for the development of drugs adapted to the individual characteristics of the patient. Polymeric micelles are nanoscale drug delivery systems ranging from 10 to 100 nm, consisting of amphiphilic conjugates of hydrophilic and hydrophobic components that self-organize in aqueous solutions, creating a two-phase structure [30,33,34,37,38,39,40]. In comparison with micelles made of low molecular weight synthetic surfactants, polymeric micelles’ main advantages can be found in their lower CMC and higher kinetic stability [34]. Indeed, polymeric micelles preserve their structure for much longer times and are non-toxic for those formed by biocompatible polymers. The positive effects of creating micellar forms of drugs are achieved due to the outer shell, which protects the drug from inactivation and stores it in a hydrophobic “fur coat”, as well as due to a decrease in the therapeutic load on the organism. In addition, it is possible to prepare gel-like formulations for children or trans-dermal cosmetic and therapeutic products [30].

Biocompatible, biodegradable chitosan and cyclodextrin were selected here as the hydrophilic part. To form micelles, these carriers are modified with oleic acid residues. Thus, the formulation is practically a dietary supplement and vitamin agent, strengthening the intestinal mucosa, antibacterial, toxin entero-sorbent, which, in addition to the main task, allows the body to receive the unsaturated essential omega-3,6,9 acids to strengthen biomembranes and repair damage in places of inflammation due to chitosan mucoadhesivity. In aspects of the effectiveness of the drug carrier, it combines mucoadhesive properties providing prolonged action with oral and inhalation use as well as enhanced biomembrane permeability (the effectiveness of the accumulation of drugs in bacterial cells). In addition, chitosan-based micelles tend to release the drug at pH 5 [41], which causes the predominant drug release in endosomes and lysosomes of macrophages [42].

The authors focused attention on spectral methods for studying the properties of micellar systems—fluorescence and FTIR spectroscopy. The properties of aromatic drug molecules depend on the microenvironment; thus, when loaded into micelles, dramatic spectral changes occur, reflected in the intensity and position of the peak of fluorescence emission and an increase in anisotropy [43,44]. The pyrene label is characterized by a three-component fluorescence spectrum, reflecting the aggregation degree of pyrene, which makes it possible to monitor micelle formation [45,46]. FTIR spectroscopy is a highly informative method since it characterizes the oscillations of analytically significant bonds, the frequencies of which change during chemical reactions, drug loading, and temperature changes—this makes it possible to monitor the inclusion of the drug molecules, determine the parameters of the phase transition, and follow the functional groups involved in the formation of supramolecular ensembles [19,24,47].

In this paper, the authors demonstrated the potential of the complex approach for the analysis of micellar systems: FTIR spectroscopy, fluorescence spectroscopy using pyrene, and drug molecules themselves (moxifloxacin/rifampicin) as labels. Modification of the CMC determination technique using pyrene is proposed: the authors used covalently labeled chitosan-OA, which allowed us to study new details of the mechanism of micelle formation. The main achievement of this work, in comparison with the literature data, is the study of the molecular mechanisms of interaction of drug molecules with micelles by changing the characteristic peaks. The FTIR spectroscopy method does not require optical transparency of the solution, so it is possible to study suspensions or gels, which is impossible using other methods.

Thus, this work is devoted to the creation of multifunctional polymer micellar systems for the treatment of bacterial infections with the possibility of expanding the fields of application, as well as methods for studying the indicated nanoparticles.

## 2. Materials and Methods

### 2.1. Reagents

Chitosan oligosaccharide lactate 5 kDa, oleic acid, 1-Ethyl-3-(3-dimethylaminopropyl) carbodiimide (EDC), 1M 2,4,6-trinitrobenzenesulfonic acid, rifampicin, moxifloxacin hydrochloride were obtained from Sigma Aldrich (St. Louis, MI, USA). Mono-(6-(1,6-hexamethylenediamine)-6-deoxy)-β-cyclodextrin (amCD) was supplied by Dayang Chem (Hangzhou, China) Co., Ltd. 1-pyrenebutanoic acid, succinimidyl ester was purchased from Invitrogen (Molecular Probes, Eugene, OR, USA). Levofloxacin (Lev, for comparison with moxifloxacin (MF)—from the class of fluoroquinolones) from Zhejiang Kangyu Pharm Co., Ltd. (Dongyang, China). Other chemicals: salts, solvents, and acids were Reakhim production (Moscow, Russia).

### 2.2. Synthesis and Characterization of Micelles

#### 2.2.1. Synthesis of OA-Grafted Chitosan and amCD

The chemical conjugates of Chit5-OA and amCD-OA were synthesized by the coupling reaction of carboxyl group of OA with amine group in the presence of 1-ethyl-3-(3-dimethylaminopropyl) carbodiimide (EDC). The oleic acid 20 mg was dissolved in 5 mL ethanol. They were then mixed at 80 °C under stirring. 10-fold molar excess of EDC and 3-fold molar excess of NHS were added into the mixture; the coupling reaction was carried out for 2 h at 70 °C under stirring. 100 mg Chit5 was dissolved in 20 mL 0.01 M HCl followed by pH increasing to 7. OA-NHS mixture was added to three portions of Chit5 to achieve a degree of modification of 5, 15, and 30% by the number of chitosan monomeric units. The mixture was stirred at 70 °C for 12 h. Synthesis of amCD-OA was carried out similarly; the molar ratio is 1 to 2 in the reagents, 1 to 1 in total. The reaction solution was then dialyzed against 50% ethanol solution using a dialysis membrane (MWCO 3.5 kDa for Chit5-OA and 1 kDa for amCD-OA) for 24 h and against water for 48 h. The degree of modification of chitosan and amCD with oleic acid was determined by the number of free amino groups in conjugates compared to the initial chitosan or amCD preparation. Kinetic curves of formation of colored adducts of chitosan amino groups with trinitrobenzenesulfonic acid (TNBS) were recorded using a UV spectrophotometer. Then, 0.05 mL of 1 M TNBS solution in water was added to a cuvette containing 3 mL of 0.1 M sodium borate buffer solution (pH 8.5), an aliquot (50 µL 1–2 mg/mL solution) of an oligo- or polymer was introduced, mixed, and a change in optical absorption at 420 nm was recorded for 60 min at 22 °C. The comparison cell contained 3 mL of the same buffer and 0.05 mL of TNBS solution. From the obtained A420 value, the content of free amino groups in conjugates was determined (in %); the content of amino groups in unmodified chitosan or amCD was taken as 100%.

#### 2.2.2. Synthesis of Pyrene-Labeled Conjugates

Covalent crosslinking of chitosan or amino-cyclodextrin amino groups with activated pyrene (1-pyrenebutanoic acid, succinimidyl ester) was carried out at pH 7.4 (0.02 M PBS) for 2 h. Molar ratio Chit:pyrene = 5:1, amCD:pyrene = 2:1

All samples were freeze-dried for two days at –60 °C (Edwards 5, BOC Edwards, Burgess Hill, UK). The degree of modification was calculated according to spectrophotometric titration of amino groups (before and after modification) with 2,4,6-trinitrobenzenesulfonic acid [25,48]. Mainly primary amino groups are detected.

#### 2.2.3. Preparation of Micelles—Critical Micelle Concentration (CMC)

Chit5-OA was dissolved in PBS (0.01 M, pH 7.4) at a concentration of 2 mg/mL. Micelle solutions were prepared by probe-type ultra-sonic treatment (50 °C, 10 min). CMC was determined using pyrene-labeled Chit5-OA and amCD-OA using the fluorescence method. The steep slope of the sigmoid corresponds to the formation of micelles (since there is a sharp change in the parameters of the fluorescence of the pyrene label, which serves as a marker for the formation of micelles). Inflection point corresponds to CMC: Analytical signal = const1/(const2 + C) + const3, where C is the concentration of Chit5-OA.

#### 2.2.4. Drug Loading into Micelles and Release

Drug (moxifloxacin, Levofloxacin, rifampicin) aqueous or ethanol solution and the polymer micellar solution were mixed and sonicated for 30 min at 50 °C. For drug loading capacity determination, analytical dialysis against distilled water for 12 h at 20 °C was performed using a dialysis membrane with 12–14 kDa cut-off with a 1:1 internal to external volume ratio. Then, the amount of antibiotics (according to A290) in the external solution and in the micelles was determined to calculate the drug loading capacity.

Then samples were freeze-dried as described above. Release of moxifloxacin was performed from the dialysis membrane (cut-off, 7 kDa, internal to external volume ratio was 1:100) at 37 °C. The amount of moxifloxacin was determined by A290. UV spectra of solutions (Section 3.3.1) were recorded on the AmerSham Biosciences UltraSpec 2100 pro device (USA) three times in the range of 200–400 nm in a quartz cell Hellma 100–QS with an optical path of 1 cm.

#### 2.2.5. Determination of the Hydrodynamic Diameter and the Degree of Aggregation of Polymer Molecules in Micelles

Determination of the hydrodynamic diameter of the synthesized polymeric micelles was carried out by nanoparticle tracking analysis (Appendix A) using the Nanosight LM10-HS device (Great Britain). Particle samples were diluted with MilliQ purified water to a particle concentration of 10^9^–10^10^ particles/mL and kept in an ultrasonic bath for 30 s. The hydrodynamic diameter was determined by the Stokes–Einstein equation due to the analysis of the trajectory of the Brownian motion of particles. Each sample was measured five times. The results are averaged and presented with a standard deviation.

#### 2.2.6. Fluorescent Micelle Visualization

Fluorescent images of micelle’s particles and aggregates were obtained using ZOE Fluorescent Cell Imager (Bio-Rad, Russia, Moscow).

### 2.3. FTIR Spectroscopy

ATR-FTIR spectra of samples’ solutions were recorded using a Bruker Tensor 27 spectrometer equipped with a liquid nitrogen-cooled MCT (mercury cadmium telluride) detector. Samples were placed in a thermostatic cell BioATR-II with ZnSe ATR element (Bruker, Ettlingen, Germany). The FTIR spectrometer was purged with a constant flow of dry air (Jun-Air, Benton Harbor, MI, USA). FTIR spectra were acquired from 900 to 3000 cm^−1^ with 1 cm^−1^ spectral resolution. For each spectrum, 50–70 scans were accumulated at 20 kHz scanning speed and averaged. Spectral data were processed using the Bruker software system Opus 8.2.28 (Bruker, Ettlingen, Germany), which includes linear blank subtraction, baseline correction, differentiation (second order, 9 smoothing points), min–max normalization, and atmosphere compensation [20,25]. If necessary, 11-point Savitsky–Golay smoothing was used to remove noise. Peaks were identified by standard Bruker picking-peak procedure.

### 2.4. Fluorescence Spectroscopy

Fluorescence of MF, Rif, and pyrene was measured using a Varian Cary Eclipse spectrofluorometer (Agilent Technologies, Santa Clara, CA, USA) at 22 °C. pH 7.4 (0.02 M PBS). Emission spectra were recorded with the following parameters: λ_exci_(MF) = 290 nm, λ_emi_(MF) = 465 nm (310–570 nm); λ_exci_(Rif) = 338 nm, λ_emi_(Rif) = 383 nm (360–570); λ_exci_(pyrene) = 340 nm, λ_emi_(pyrene) = 377, 397, and 467 nm (360–600). The fluorescence anisotropy was determined using four components at different positions of the manual polarizer.

### 2.5. Mathematical Equations and Calculations

(1)The critical concentration of micelle destruction is determined based on the intensity of the peaks of fluorescence of pyrene attached to chitosan amino groups. Peaks 377 and 397 nm are extinguished, and 467 nm increases with the dissociation of the micelles. The coordinate along the abscissa axis corresponding to the inflection point of the curve is CMC.(2)The degree of inclusion of drugs by weight is determined by the method of analytical equilibrium dialysis (cut-off 7 or 12–14 kDa).

### 2.6. Antibacterial Activity of Moxifloxacin and Levofloxacin in Micelles

The strain used in this study was *Escherichia coli* JM109 (J.Messing, Minneapolis, MN, USA). The culture was cultivated for 18–20 h at 37 °C to CFU ≈ 1.5 × 10^8^–2 × 10^8^ (colony-forming unit, determined by A600) in liquid nutrient medium Luria–Bertani (pH 7.2) without stirring. The experiments in liquid media were conducted by adding 50 μL of the samples to the 5000 μL of cell culture. The specimens were incubated at 37 °C for seven days. At the specific time, 100 μL of each sample was taken, diluted with distilled water, and the absorbance was measured at 600 nm. For quantitative analysis of the dependences of CFU (cell viability) on the concentration of MF or Lev, 50 μL of each sample was diluted 10^6^–10^8^ times and seeded on the Petri dish. Dishes were placed in the incubator at 37 °C for 24 h. Then the number of the colonies (CFU) was counted.

### 2.7. In Vivo Experiments

#### 2.7.1. Animals

Animals (rats) starved 12 h before the experiment, leaving free access to water. The mass of the animals was (400 ± 40) g. The animals were divided into groups of 3 in each. Solutions of the studied substances in the volume of 1 mL were administered orally using a probe. Considering the concentration and volume of the administered drugs, as well as the weight of the animals, the administered dose was 12.5 mg/kg. Statistical analysis of obtained data was carried out using the Student’s *t*-test Origin 2022 software (OriginPro 2021 9.8.0, OriginLab Corporation, Northampton, MA, USA). Values are presented as the mean ± SD of three experiments.

#### 2.7.2. Protocol of Experiments on the Study of MF Pharmacokinetics

Solutions for oral administration were prepared at concentrations of 5 mg/mL converted to MF. From the caudal vein through 15, 30, 60, 90, 120, 180, 240, 360 min, 24, 48, and 72 h, 100–150 µL of blood were taken into tubes with the anticoagulant EDTA. For MF extraction, 450 mL of methanol with 0.05 μg/mL of Lev (internal standard) was added drop by drop to a 50 mL blood aliquot. The obtained extracts were centrifuged at 4000 g for 10 min on an Eppendorf R5810 centrifuge to precipitate proteins. The concentration of MF was determined in the supernatant on an Agilent 1100 Series chromatograph with an LC/MS G1956B mass spectrometric detector (Agilent Technologies, Santa Clara, CA, USA). The calculation of pharmacokinetic parameters was carried out on the basis of the obtained values of the MF concentration at different points in time after oral administration using the Borgia program (version 1.03, Copyright 1999–2000, Nauka Plus).

## 3. Results and Discussion

### 3.1. Synthesis and Characterization of Polymeric Micelles

The synthesis of chitosan and cyclodextrin derivatives was carried out by activating the carboxyl group OA using EDC and NHS (to form a stable intermediate OA-NHS), followed by crosslinking with amino groups Chit or amCD—the scheme is shown in Figure 1. Fluorescent image of Chit5-OA-5 micelles, labeled with doxorubucin. The photos show nanoparticles (~200 nm) and their aggregates up to 10 microns. Therefore, micelles are formed and include a dye, so we can observe fluorescent particles.

The chemical structure was studied using FTIR spectroscopy (Figure 2). ATR-FTIR revealed a number of intensive bands of a typical lipids particle spectrum: they comprise bands of asymmetric and symmetric valence oscillations of methylene groups CH_2_ (as) and CH_2_ (s) (2915–2930 cm^−1^ and 2850–2855 cm^−1^, respectively), which are sensitive to changes of lipid package in the supramolecular structure. The FTIR spectra of studied compounds (Chit5, amCD, and OA) show characteristic peaks corresponding to valence oscillations of C-H bonds, which undergo a strong shift during conjugation from 2982 to 2958 cm^−1^, from 2924–2925 to 2917 and from 2857 to 2850 cm^−1^. Such a decrease in the positions of the CH_2_ peaks corresponds to a transition to a more structured (decrease in the fluctuation degree) pseudo-gel-like state with a dense lipid package [20]. The formation of a covalent bond between oleic acid and chitosan is confirmed by a decrease in the intensity of the peak of the COOH group (1730–1770 cm^−1^) of OA and the appearance of 1550–1580 (amide –NH–) and 1600–1710 cm^−1^ (–C(=O)–). Oscillations in the acetyl group of chitosan are “quenched” and make a small contribution. Bands of OA carbonyl groups (Figure 2, 1720–1750 cm^−1^) have multicomponents [20,24,49,50]. High-frequency components correspond to low-hydrated groups, and low-frequency components correspond to highly hydrated groups. A decrease in hydration degree can be expected upon micelles formation. An increase in hydration degree can be observed during melting (transition to liquid-crystal state) due to the higher availability of carbonyl groups to aqueous molecules. For the considered grafted chitosans and cyclodextrin, the shift of the carbonyl absorption band to the high-frequency region and the decrease in the positions of the CH_2_ groups of oleic acid residues (Figure 2) confirms the formation of micellar systems.

Based on FTIR spectroscopy data (integral peak intensities), nanoparticle tracking analysis (Appendix A), and initial component ratios, the average component ratios in conjugates were determined (Table 1). Micelles consist of modified amCD and OA with a molecular weight of 1.5 kDa and 5.5–7.5 kDa, respectively. The degree of OA grafting in chitosan varied from 5 to 30%. Only one OA residue is attached to the amCD, so that there are no steric difficulties during micelle formation. Using the NTA (nanoparticle tracking analysis) method based on the analysis of light scattering data, the degree of aggregation of surfactant molecules in micelles was determined. At a total polymer concentration of 0.2 μg/mL (taking into account molar mass (MM) of polymeric molecules equal to 30–130 pM) the number of micellar particles was estimated as (from 7 to 39 for different samples) × 10^9^ particles/mL, which means that the micelles composition is approximately 1200–2000 polymer chains per one micelle. An increase in the degree of chitosan modification to a certain value, according to literature data [27,28,29,31,34,41,51,52,53] (up to 30–40%), improves micelle formation due to the compaction of the hydrophobic core and, consequently, thermodynamically more preferable inclusion of aromatic drugs. A small degree of modification causes a loose core and a larger micelle size.

### 3.2. Critical Micelle Concentration

Critical micelle concentrations were determined by fluorescence spectroscopy using pyrene label covalently conjugated to chitosan or amCD NH_2_-group (through butyl spacer (amide crosslinking—Figure 1b)). The literature describes pyrene monomer-excimer techniques based on changing the intensity ratio I3/I1 of pyrene peaks (Figure 3, I1—377 nm, I2—397 nm, I3—467 nm) during the formation/destruction of micelles or phase transitions [45,54]. The first two peaks correspond to the monomer. The third wide peak (467 nm) corresponds to an excimer—an excited dimer. It was found that in solutions with a polymer concentration below CMC, the main form of existence of the pyrene is the excimer, since hydrophobic pyrene molecules attached to chitosan form stacking interactions in the non-regular polymeric system. At a concentration of polymers above CMC is where micelles are formed, and the monomeric pyrene form prevails in the system since the pyrene attached to chitosan through the spacer is immersed in the hydrophobic part of the micelle and stacking interaction is suppressed due to shielding with fat tails of OA. These data are confirmed by pyrene spectra in water–ethanol and octane (Figure 3a,d): pyrene in octane is a monomer; pyrene in hydrophilic media forms aggregates existing mainly in the excimer state. Confirmation of this hypothesis is provided by the fluorescence anisotropy spectra (Figure 3b,e, Appendix A). In dilute polymer solutions, the anisotropy value is 0.1–0.25 units, which corresponds to aggregated forms in disordered polymer branches of chitosan labeled with pyrene. In solutions with concentrations above CMC, due to the ordering of amphiphilic molecules and the formation of micelles, the fluorescence anisotropy decreases due to the dissociation of pyrene-containing aggregates and the formation of regular micellar structures. The curves of the dependence of the fluorescence intensity of the three peaks are shown in Figure 3c,f, Appendix A: CMC approximately corresponds to the intersection point of the increasing and decreasing curves. The CMC are presented in Table 1. For the formation of micelles, a lower concentration is required in the case of 15 and 30%-modified chitosan compared with amCD-OA and Chit5-OA-5.

### 3.3. Loading of Moxifloxacin and Rifampicin into Polymeric Micelles

The loading capacity of the micelles by MF was determined by analytical equilibrium dialysis with UV detection after ultrasound treatment. Table 1 shows the quantitative parameters of loading by weight—about 40% for MF and 24% for Rif.

#### 3.3.1. FTIR Spectroscopy

The advantage of FTIR spectroscopy is that the method provides information about the microenvironment of characteristic bonds (e.g., aromatic systems) and the molecular details of the interaction of the drug molecule with the delivery system [19]. FTIR spectra of both free MF, Rif, and its micelle form with different loading degrees are presented in Figure 4 and Appendix A. For moxifloxacin in the FTIR spectrum, the most significant are the valence oscillations of the C–H bonds of the aromatic system 1450 cm^−1^ and aliphatic C–H 2917 and 2850 cm^−1^ (Figure 4a) [55,56,57]. When MF is loaded into micelles, the oscillations of aromatic C–H shift towards large wavenumbers due to a change in the MF microenvironment to a more hydrophobic one (Figure 4b). At the same time, the environment of oleic acid residues in polymer changed dramatically, indicating the efficiency, including MF in micelles. It should be mentioned that the peak position of CH_2_ asymmetric valence oscillation mainly depends on the mobility of acyl chains: for gel-like state ν_as_(CH_2_) is around 2917–2920 cm^−1^, while for the fluid-like state, this value reaches 2925 cm^−1^. So, when the lipid system becomes more disordered, the peak shifts to higher frequencies and vice-versa. Upon the introduction of MF into the micelles—shifts of CH_2_ oscillations towards smaller wave numbers from 2923 and 2851 to 2916 and 2849 cm^−1^ are observed (Figure 4b), indicating the stabilization of the gel-like state of the micelles. MF molecules are incorporated into micelles with a protonated form of COOH groups (Figure 4a and Appendix A): the peak at 1720 cm^−1^ corresponding to the carboxyl group of the drug molecule is extinguished in the FTIR spectra. This proves the inclusion of MF in the hydrophobic cavity since the charged molecule cannot penetrate into the hydrophobic cavity. A similar pattern was observed when MF was incorporated into the hydrophobic cavity of cyclodextrin as part of polymer conjugates. It should be mentioned that the inclusion of MF in liposomes also stabilizes the gel-like state of lipids and increases the temperature of the phase transition [50].

The data on the quantitative content of MF and Rif (Table 1) on release by analytical dialysis are confirmed by shifts of the components in the FTIR spectrum to the region of large wavenumbers (1441.8 to 1445.8 cm^−1^, 1453.2 to 1456.1 cm^−1^), which corresponds to the inclusion of the drug in the hydrophobic core of micelles and a change in the microenvironment. Moreover, in the micellar MF spectrum, a third component can be explicitly distinguished (1463.5 cm^−1^—corresponds to the loaded drug), which makes it possible to determine the content of MF or Rif in the cores. When MF is included in the micelles, the integral ratio of the hydrophobic component (1456 and 1463 cm^−1^) to the hydrophilic component (1441 cm^−1^) increases from 0.36 to 0.80—Figure 4c. Which corresponds to about 30–40% of the drug load—close to the TNS titration data.

The inclusion of rifampicin in the composition of polymeric micelles is accompanied by an increase in the intensity of peaks of aromatic C–H oscillations, while the intensity of peaks corresponding to oscillations of aliphatic CH_2_ in OA residues decreases, and their maximum shifts towards smaller wavenumbers (Figure 4d), what also causes ordering in micelles is the filling for micelles. Similar low-frequency shifts are described in work [49] for rapamycin when loaded into liposomes. Rapamycin, a large hydrophobic molecule, was actively embedded in the bilayer of liposomes, which led to the ordering of acyl chains [49]. C–O–C bonds in chitosan (1080 cm^−1^) are also sensitive to micelle formation (Figure 4). The intensity of the absorption band 1080 cm^−1^ increases to a much lesser extent relative to those values that would be expected when taking into account the mass excess. This is due to the fact that with an increase in the polymer concentration, the molecules are ordered, and micelles are formed with subsequent compactification of polymer chains, which leads to a decrease in the absorption coefficient of C–O–C bonds.

#### 3.3.2. Fluorescence Spectroscopy of MF and Rif upon Incorporation into the Micelles

The fluorescence depends on the microenvironment, the concentration of molecules, and pH—therefore, when loading drugs with fluorescent properties into delivery systems, changes in the intensity and position of the maximum fluorescence emission are expected, as well as changes in anisotropy correlating with molecular mobility and the degree of aggregation of molecules. Figure 5a shows the emission spectra of MF fluorescence in free form and loaded into micelles. When incorporated into a hydrophobic “coat” of OA residues, the intensity of MF fluorescence decreases while the maximum shifts to the long-wave region (Figure 5b). An increase in the fluorescence anisotropy (Figure 5c), correlating with a decrease in the molecular mobility of the fluorophore, confirms the penetration of MF into micelles. Similar changes are observed for rifampicin, except for changes in intensity (Figure 5d). An increase in the intensity of Rif fluorescence indicates an increased solubility of this hydrophobic drug and the destruction of rifampicin microcrystals. Thus, loading drugs into micelles allows them to increase their solubility and protect them from destruction.

The resulting data on the loading of drugs into micelles determined by the dependences of the peak positions in the FTIR spectra and peak position, intensity, and anisotropy in fluorescence spectra are shown in Table 2. The best results were demonstrated by micelles based on Chit5-OA-30; due to the larger number of fatty residues, denser micelles are formed, and the drug is held more firmly. Cyclodextrin-based micelles additionally incorporate the drug into the cyclodextrin hydrophobic cavity, which further improves the properties of the delivery system. All the micelles studied have demonstrated significant potential in terms of loading therapeutic agents.

An interesting fact has been discovered. The fluorescence anisotropy was also studied by a pyrene label (Section 3.2), but the pyrene label was covalently attached via a butyl spacer to a chitosan. So, in the case of pyrene conjugate, anisotropy of chitosan was an object for observation: the formation of ordered structures of the copolymer from aggregates during micelle formation. In the case of free drugs (and fluorophores at the same time), anisotropy increases due to a decrease in molecular mobility of low molecular drugs when MF or Rif is incorporated into micelles.

### 3.4. Phase Transitions of Micelles with the Drug

The study of phase transitions is important from the point of view of the thermodynamic stability of micelles, and, in addition, it determines the drug release properties. Since the particles tend to release the drug mainly during the phase transition, it is desirable that the phase transition occurs just at 37–40 °C. Figure 6a shows the fluorescence spectra of MF loaded into polymeric micelles, depending on temperature. Figure 6b shows the dependence of the intensity and position of the maximum fluorescence of MF on the temperature–phase diagrams. When the micelles loaded with the drug are heated, due to the increase in the OA residues mobility and the possibility of implementing hydrophobic interactions of chitosan (to a lesser extent), phase transitions occur at 42–43 °C. Further, 30%-grafted chitosan forms are less prone to gelation micelles than in the case of 15 and 5% modifications (where chitosan is the main component); therefore, the sensitivity to temperature increases and the phase transition starts at a lower temperature: 30–40 °C—slow phase, above 40 °C—rapid release. amCD micelles undergo transitions at 45 °C due to the smaller size and greater hydrophilicity of cyclodextrin relative to chitosan; therefore, Chit5-OA-30 micelles are the most optimal, since they show high parameters of drug loading and, in addition, the phase transition temperature is optimal for creating dosage forms. We will pay special attention to the possibility of regulating the release rate of MF—which will create stimulus-sensitive systems for adaptive therapy.

### 3.5. Moxifloxacin In Vitro Release from Polymeric Micelles

Prolonged drug release is the possibility of creating formulations with a maintenance concentration of the antibiotic inside the therapeutic window for a long time, which will significantly increase the effectiveness of treatment and probably combat multidrug resistance. Previously, the prolonged release of fluoroquinolones from polymer particles was shown [25,34,58]. Figure 7 shows the curves of MF release through a dialysis membrane from MF-loaded micelles. Chitosan-based micelles are more effective due to the additional retention of MF by polymer chains. The half-release period of the antibiotic in a simple form is 0.7 h, and from the micelles amCD-OA—1.6 h, Chit5-OA—from 25 to 110 h. The release period of the main part of the drug (80%) has been increased from 4h to 100–200 h. An additional advantage is achieved due to the low initial accumulation rate—so that the MF pharmacokinetic concentration profile is not a sharp peak, but a gentle curve.

Table 3 shows the periods of half-release of MF from polymer systems, free MF, and unmodified chitosan particles. Micelles, in comparison with unmodified chitosans and cyclodextrins, slow down the rate of initial release (tgα = release % per hour) from 100–120 to 10 for the leading sample. Article [59] shows that the β-cyclodextrin-MF complex slows the release of MF by up to 60–80% (for 100 min). However, this effect is only marginal since, in micellar systems, the deceleration of the half-release period can reach two orders of magnitude over time. Micelles release the drug due to local defects and mobility of polymer heads-chitosans. The release rate decreases with an increase in the degree of chitosan modification due to the compaction of the inner core of the micelle. Data on the release of rifampicin are described in the literature for similar chitosan-based polymer micelles [53] and presented in Figure 7b. The effect of Rif inclusion in Chit5-OA-15 micelles and analogs is clearly noticeable: the initial release rate of the drug is slowed down by more than two times, and the half-release period is increased from 1 to 16 h. The initial release rate of MF and Rif is comparable, but a further release of MF is slower: in terms of the half-release period by 2–3 times.

### 3.6. Antibacterial Activity of Fluoroquinolones in Polymeric Micelles

Molecular containers based on polymers and cyclodextrins enhance the effect of the antibiotic by preventing destruction in biological fluids, which is shown on the model medium in a number of works [19,25,60]. In addition, polymer systems cause increased penetration of the drug into the cells due to adsorption on the bacterial wall [47,58]. Figure 8 shows the curves of the survival of *E. coli* bacterial cells on the incubation time with various forms of moxi- and levofloxacin (Lev was used to demonstrate the applicability of synthesized polymers to various medicinal molecules, including from a number of fluoroquinolones). Apparently, polymeric micelles accelerate the penetration of antibiotics into cells due to the adsorption of polysaccharides on the cell wall surface, which is reflected in the initial rate of reduced cell viability: MF in free form killed 50% of bacteria (relative to the control cells) after 2 h and after 1.4 h in the micellar form; in the case of Lev 2.2 and 0.8 h, respectively. Enhanced membrane permeability makes the antibacterial agent more effective, since free forms of Lev and MF at a concentration of 5 μg/mL did not exceed cell viability by 18–20%, while in micelles, antibiotics killed almost 100% bacteria after 2 h. The prolongation of the action of the antibiotic (right sections of the curves) is achieved by preserving part of the molecules in the loaded form, in a micelle coating. Previously, the authors showed the effect of the destruction of antibiotics in the cellular environment, as opposed to antibacterial agents, in the composition of polymer systems [19,25,60]. In addition, the importance of using polymeric micelles is shown (Figure 8) since they increase the penetration into cells and the effectiveness of the antibiotic. For comparison: simple cyclodextrin has a negligible effect on the antibiotic.

### 3.7. Pharmacokinetics of MF in Polymeric Micelles

Figure 9 shows the pharmacokinetic curves of moxifloxacin in free form and loaded into chitosan-based particles in comparison with the control system—monomeric CD. With oral administration, the maximum concentration of MF, which is achieved in the blood, increases by two times in micellar systems, and the effective concentration (integral of the curve, AUC) increases by 1.5–2 times. Why is it necessary to use complex polymeric micelles? MF in β-cyclodextrin was selected as a control sample: pharmacokinetic parameters improved by no more than 5–10% (within the margin of error), and micelles allowed us to achieve a several-fold enhancement of effects. For the control system—monomeric CD (without micelles), we do not observe prolonged pharmacokinetics since the drug–CD binding constants do not exceed 10^3^–10^4^ M^−1^, so the drug is released very quickly. Further, cyclodextrin itself has a low molecular weight to significantly affect the time of action. Thus, the developed micelles are promising in the aspects of creating enhanced forms of antibacterial drugs.

### 3.8. Comparison of Micelles with Existing Systems—Advantages and Disadvantages

Table 4 shows the comparative characteristics of micelles developed by the authors and described in the literature. Chitosan-based micelles are superior to micelles based on other components, for example, based on poly(d, l-lactide-co-glycolide), inorganic materials. Comparing chitosan-based micelles with similar systems in works [27,36], we see the advantages of our conjugates: a high degree of modification by fatty acids, and as a result, a smaller CMC, a higher packing density of fatty tails, and a smaller size of micelles. In addition, our systems are characterized by high drug loading capacity (35–48% vs. 13–20), and increased drug release time. The pharmacokinetic parameters of MF were increased comparably with the data for docetaxel in work [36]. Thus, the developed micellar systems have great therapeutic potential in comparison with existing platforms.

It is worth noting that the use of polymer micelles has a number of possible limitations: (1) the occurrence of an immune response to the introduction of foreign particles (a possible solution is the use of chitosan oligosaccharides of 5 kDa, as in this work), (2) thrombogenicity and/or hemolysis of erythrocytes (not typical in most cases for biocompatible micelles such as ours), (3) nonspecific accumulation in tissues (the solution is the use of an address vector, for example, a trimannoside vector on receptors CD206 macrophages), (4) effect on the hormonal background of the body, (5) effect on lipid metabolism. These factors should be kept in mind during preclinical and clinical trials.

## 4. Conclusions

This work is devoted to the creation of polymeric micelles based on chitosan and cyclodextrin, grafted with oleic acid residues. These molecular containers are applicable to improve the properties of antibacterial and, in the future, antitumor drugs. The average size of micelles is 70–120 nm, which is optimal for the prolongation of the drug’s action and, at the same time, the absence of an immune response and thrombogenicity. CMC—4–30 nM was determined using fluorescence spectroscopy using a pyrene label, and it was shown that the loading of antibiotics moxifloxacin and rifampicin was up to 40% by weight. Molecular details of the interaction of aromatic medicinal agents with the hydrophobic core of micelles were studied using FTIR spectroscopy. The authors paid special attention to the thermal sensitivity of micelles-phase transitions. The most prospective polymer turned out to be Chit5-OA-30, which releases the drug slowly at 32–38 °C, and at higher temperatures, the release rate increases significantly, which will create adaptive delivery systems and controlled release of the drug. It has been shown to increase antibacterial activity on *E. coli* cells and maintain antibiotic activity for more than 4 days. The pharmacokinetic parameters of MF included in micelles are two times higher than those for simple MF and loaded into cyclodextrin. Thus, we have demonstrated the significant therapeutic potential of polymeric micelles and their prospects for enhancing antibacterial drugs: the minimum inhibitory concentration is reduced due to an increase in the drug solubility and bioavailability.

## Figures and Tables

**Figure 1 life-13-00446-f001:**
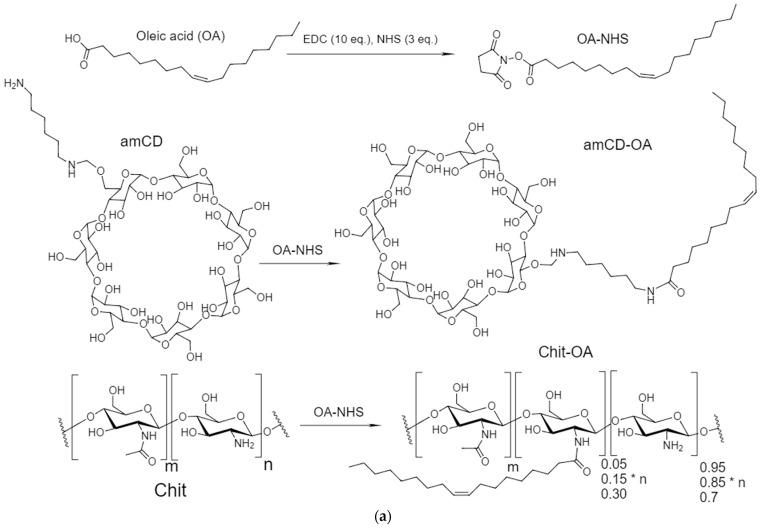
(**a**) Scheme of synthesis of chitosan (Chit) or cyclodextrin (amCD) conjugates with oleic acid (OA). (**b**) Schematic representation of polymeric micelles. (**c**) Fluorescent image of Chit5-OA-5 micelles, labeled with doxorubucin.

**Figure 2 life-13-00446-f002:**
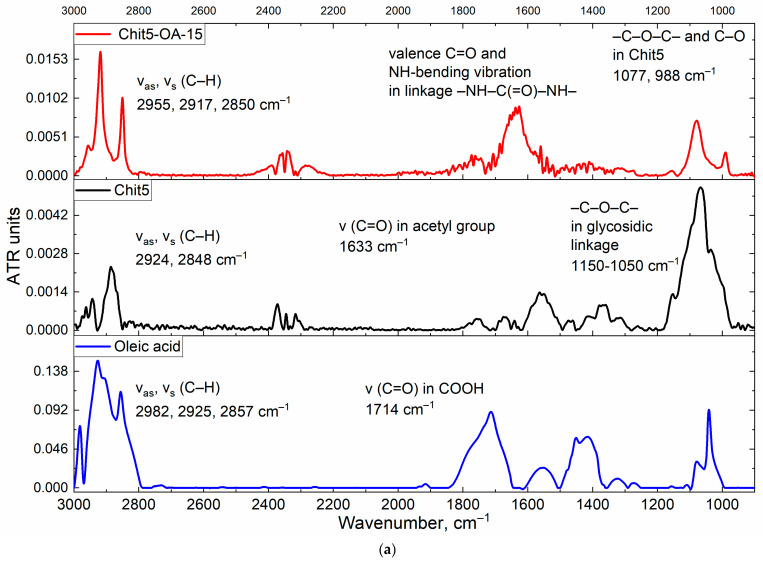
(**a**) FTIR spectra of Chit5-OA-15 grafted polymer in H_2_O; Chit5 in H_2_O; oleic acid in EtOH. (**b**) FTIR spectra of amCD-OA in H_2_O; amCD in H_2_O; oleic acid in EtOH. Background spectra were subtracted. T = 22 °C.

**Figure 3 life-13-00446-f003:**
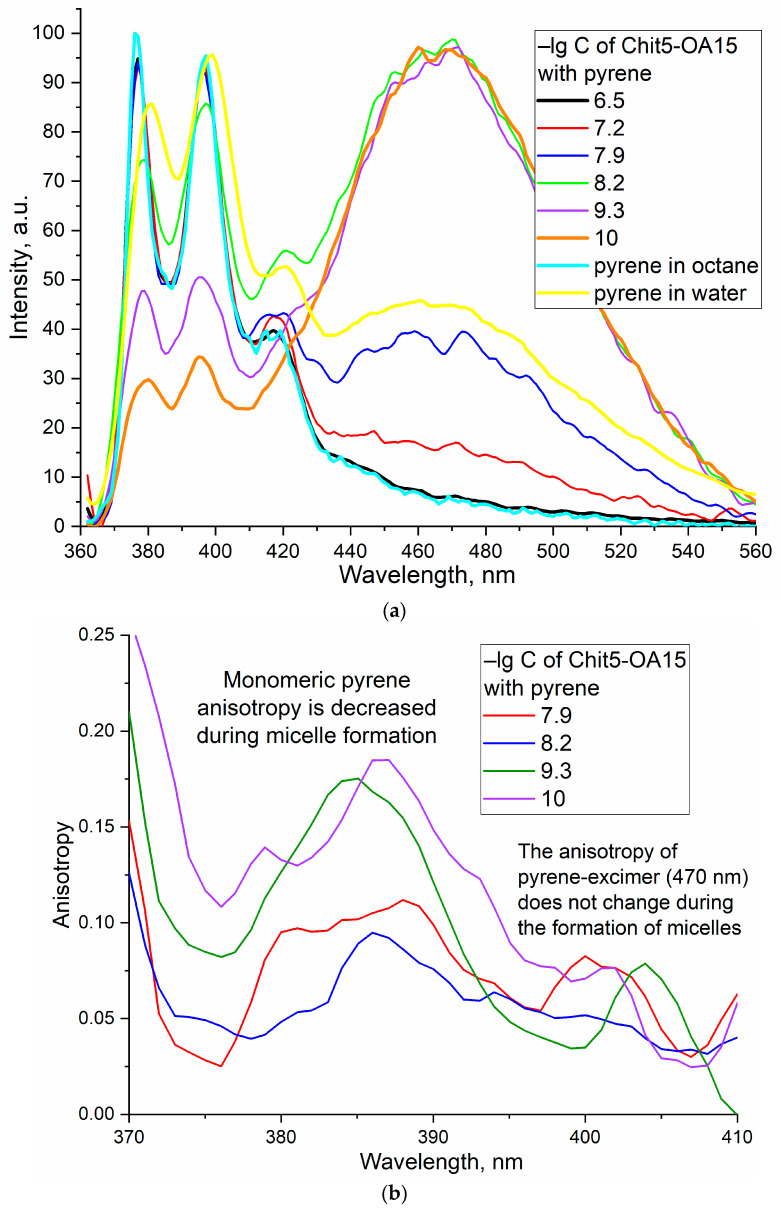
Fluorescence determination of the critical concentration of micelle formation of Chit5-OA-5. (**a**) Normalized fluorescence emission spectra of pyrene label. (**b**) Pyrene fluorescence anisotropy spectra. (**c**) Peaks intensities of the spectra (**a**) on the concentration of Chit5-OA-5 dependences. The analytical signal—the intensity of the three peaks of pyrene fluorescence, as well as the ratio of the third and first—correlates with the formation of micelles. The steep slope of the sigmoid corresponds to the formation of micelles. Inflection point—approximately corresponds to CMC. (**d**–**f**) similar to (**a**–**c**) for amCD-OA. PBS (0.01 M, pH 7.4). λexci(pyrene) = 340 nm. T = 22 °C.

**Figure 4 life-13-00446-f004:**
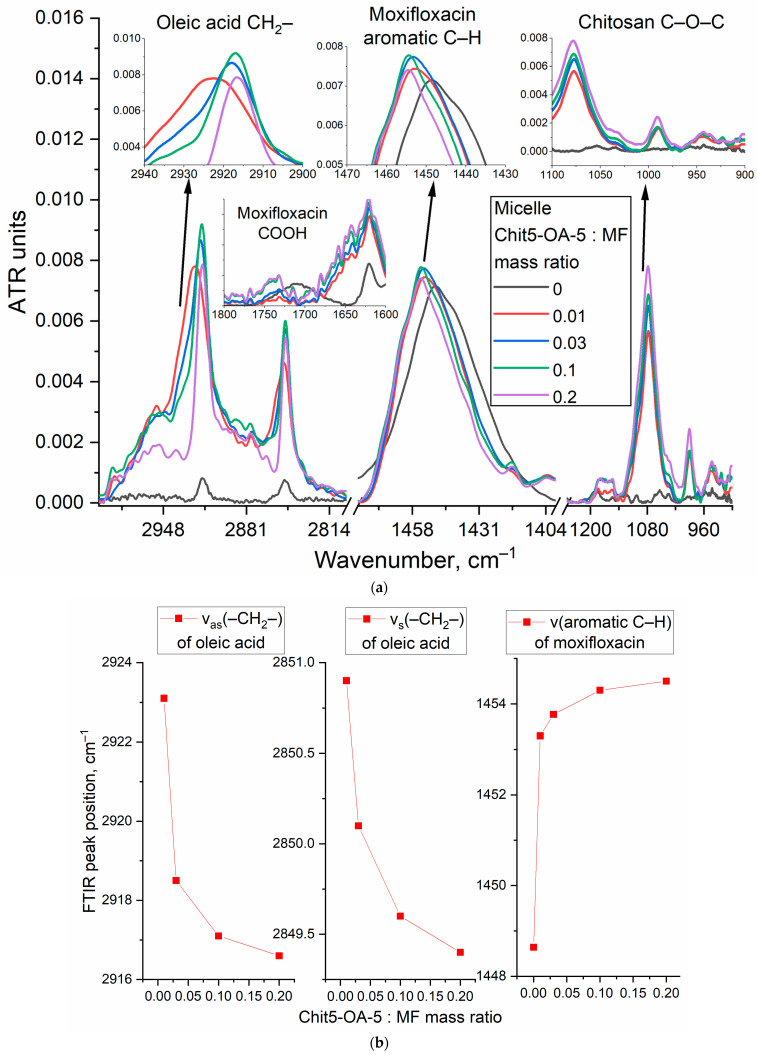
(**a**) FTIR spectra of moxifloxacin (MF) and its inclusion form into Chit5-OA-5 micelles. (**b**) MF FTIR peak position vs. micelle excess—dependence on the inclusion degree. (**c**) MF FTIR spectra deconvolution: in free form and loaded into micelles Chit5-OA-5. (**d**) FTIR spectra of rifampicin (Rif) and its inclusion form into Chit5-OA-15 micelles. 0.01M PBS (pH 7.4). T = 22 °C.

**Figure 5 life-13-00446-f005:**
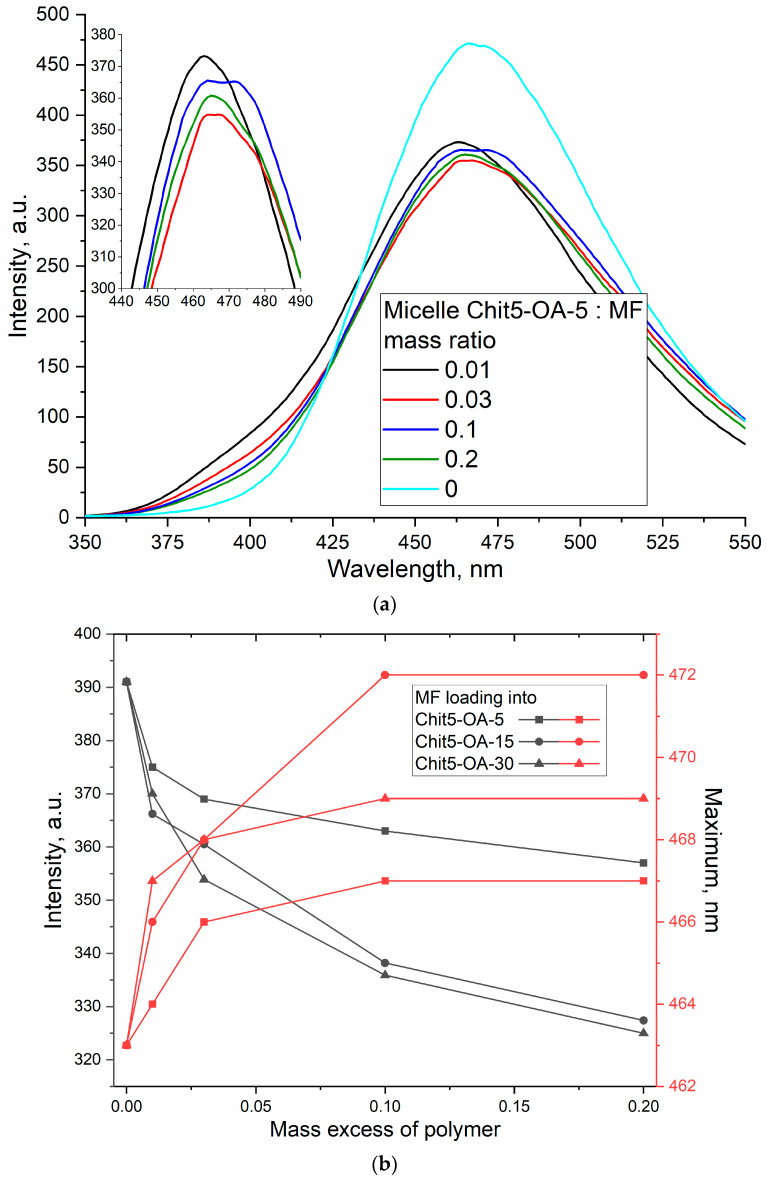
(**a**) Fluorescence emission spectra of MF and its Chit5-OA-5 micelle formulation. (**b**) The dependencies of the maximum position and intensity of MF fluorescence on the degree of loading into the micelles. (**c**) The dependences of the MF fluorescence anisotropy on the degree of loading into micelles. (**d**) Fluorescence emission spectra of Rif and its Chit5-OA-15 micelle formulation. T = 22 °C. 0.01M PBS (pH 7.4). λ_exci_(MF) = 290 nm, λ_exci_(Rif) = 338 nm.

**Figure 6 life-13-00446-f006:**
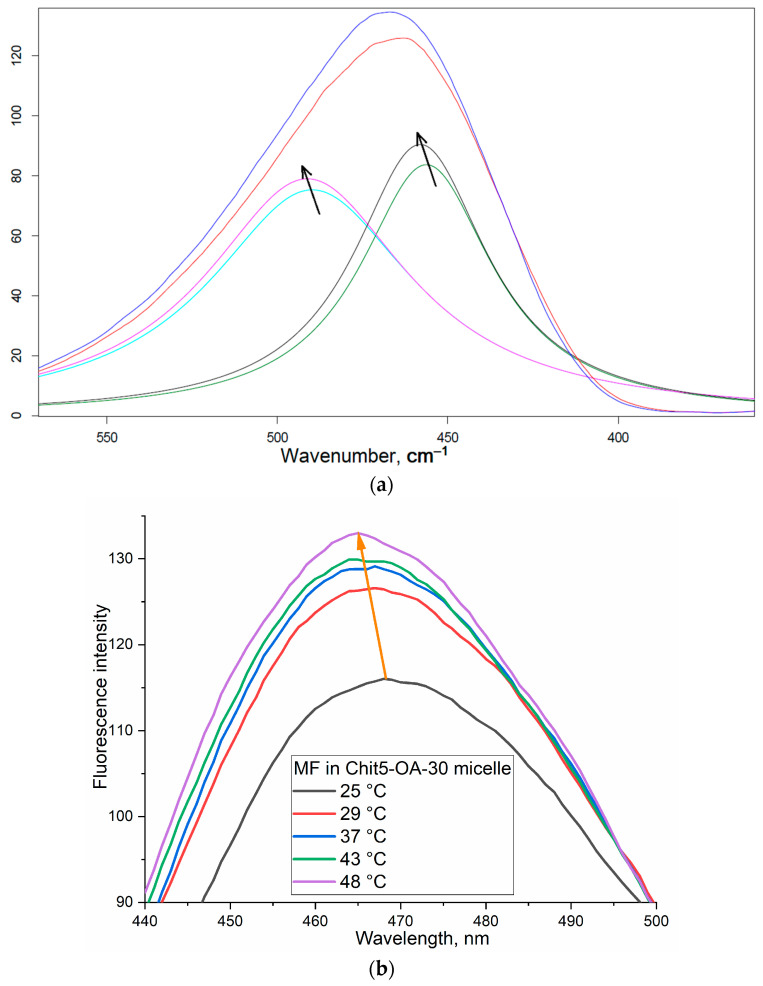
(**a**) Fluorescence spectra of MF loaded into polymeric micelles Chit5-OA-5 at 25 (blue) and 47 °C (red), deconvoluted into two gaussian components. (**b**) Fluorescence spectra of MF loaded into polymeric micelles Chit5-OA-30 at 25–47 °C. (**c**) The dependencies of the intensity and position of the maximum MF fluorescence–phase diagrams. 0.01 M PBS, pH 7.4.

**Figure 7 life-13-00446-f007:**
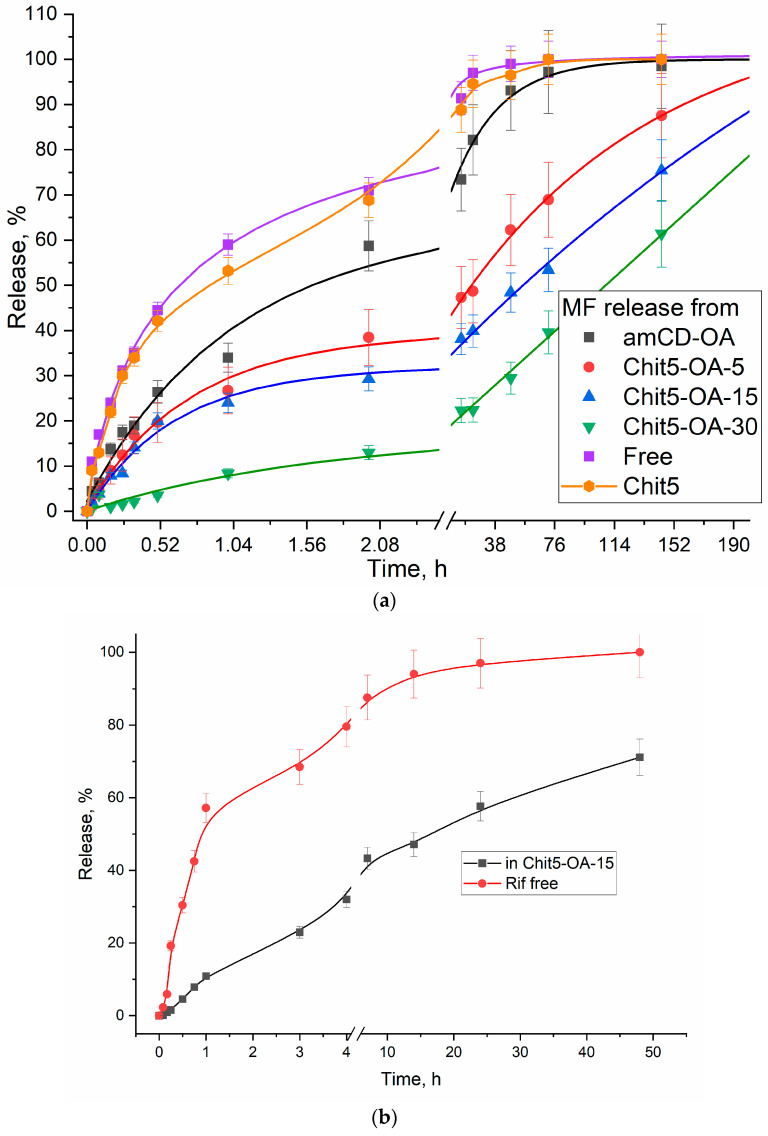
Release kinetics curves of: (**a**) free moxifloxacin and micellar form of moxifloxacin complexes (2 mg/mL, 40% MF by mass), (**b**) free rifampicin and micellar form of rifampicin complexes (0.5 mg/mL, 15% Rif by mass). Dialysis membrane (7 kDa cut-off) into an external solution (1:5 by volumes). PBS (pH = 7.4). Moxifloxacin was detected by UV absorption at 290 nm. Rifampicin was detected at 470 nm. T = 37 °C.

**Figure 8 life-13-00446-f008:**
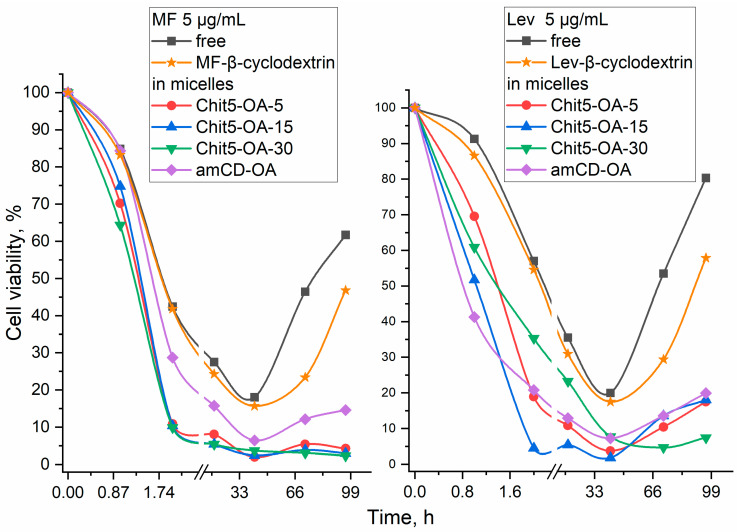
Cell viability of *E. coli* (JM109) as a function of incubation time with drug samples. Major active component is MF or Lev 5 μg/mL in free form and loaded into polymeric micelles Chit5-OA5, 15, 30, and amCD-OA. CFU (0 h) = 3 × 10^6^. pH 7.4 (0.01 M PBS), 37 °C.

**Figure 9 life-13-00446-f009:**
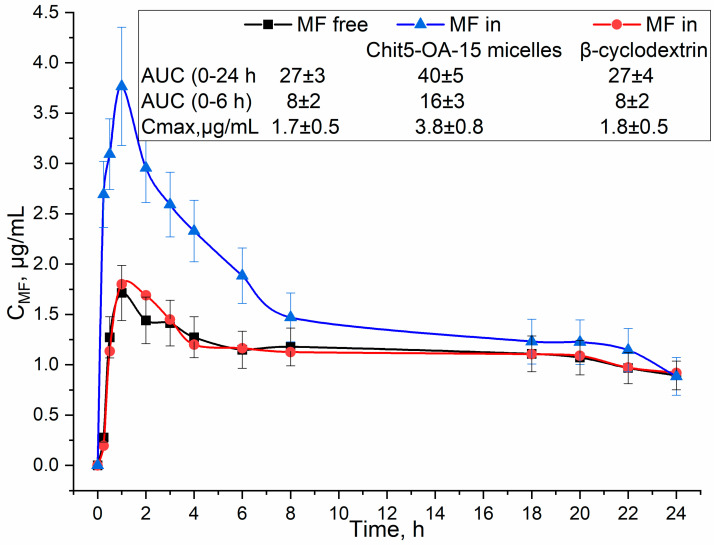
Pharmacokinetic curves of MF and its formulations with β-cyclodextrin (control), and chitosan polymeric micelles in Wistar rats. Oral administration of 12.5 mg/kg MF. AUC—area under the curve.

**Table 1 life-13-00446-t001:** Polymeric micelle’s physico-chemical characteristics. T = 22 °C.

Micelle Designation	Molecular Weight of One Structure Unit, kDa	Hydrodynamic Diameter **, nm	Critical Micelle Concentration, nM	Loading Capacity by MF, Mass %
Chit5-OA-5 *	5.5 ± 0.3	average 112 ± 67; major 45–80; 100–110	32 ± 3	40 ± 3
Chit5-OA-15 *	6.3 ± 0.4	average 100 ± 52; major 50–70; 85–105	5 ± 1	44 ± 1
Chit5-OA-30 *	7.5 ± 0.6	average 89 ± 37; major 50–80	4 ± 1	48 ± 3
amCD-OA	1.5 ± 0.1	average 132 ± 63; major 60–100; 140–160	20 ± 6	35 ± 2

* ChitX-OA-Y, where X = 5 kDa, molecular weight of chitosan, Y–OA: (monomers of chitosan), modification degree, determined by TNBS spectrophotometric titration of chitosan, amCD and conjugates NH_2_-group (Appendix A) and FTIR deconvolution; ** by NTA, Nanoparticle tracking analysis (Appendix A).

**Table 2 life-13-00446-t002:** Parameters of MF and Rif loading into polymeric micelles. Based on the dependence curves of the peak positions in the FTIR spectra and peak position, intensity, and anisotropy in fluorescence spectra. The number of polymers to achieve 50 and 90% of the drug load. T = 22 °C.

Micelle Designation	Weight Ratio of Polymer: MF		Weight Ratio of Polymer: Rif	
	for 50% MF Loading	for 90% MF Loading	for 50% Rif Loading	for 90% Rif Loading
Chit5-OA-5 *	0.07 ± 0.02	0.34 ± 0.05	0.18 ± 0.06	0.9 ± 0.1
Chit5-OA-15 *	0.07 ± 0.02	0.29 ± 0.06	0.17 ± 0.06	0.8 ± 0.1
Chit5-OA-30 *	0.06 ± 0.01	0.25 ± 0.04	0.11 ± 0.02	0.65 ± 0.05
amCD-OA	0.06 ± 0.02	0.25 ± 0.03	0.12 ± 0.03	0.7 ± 0.1

* ChitX-OA-Y, where X = 5 kDa, molecular weight of chitosan, Y–OA: (monomers of chitosan), modification degree, determined by TNBS spectrophotometric titration of chitosan, amCD and conjugates NH_2_-group (Appendix A) and FTIR deconvolution.

**Table 3 life-13-00446-t003:** The half-release period and initial release rates of MF. The conditions are similar to those given in Figure 7.

MF Form	Half-Release Period τ_1/2_, h	Initial Release Rate (tgα)
Free	0.65 ± 0.07	120 ± 10
Chit5	0.8 ± 0.1	110 ± 8
micelle Chit5-OA-5 *	23 ± 2	40 ± 5
micelle Chit5-OA-15 *	56 ± 7	35 ± 3
micelle Chit5-OA-30 *	108 ± 12	9.2 ± 0.7
micelle amCD-OA	1.6 ± 0.2	53 ± 4

* ChitX-OA-Y, where X = 5 kDa, molecular weight of chitosan, Y–OA: (monomers of chitosan), modification degree, determined by TNBS spectrophotometric titration of chitosan, amCD and conjugates NH_2_-group (Appendix A) and FTIR deconvolution.

**Table 4 life-13-00446-t004:** Comparison of the parameters of the developed micelles with those described in the literature.

References	Micelle’s Designation *	Chitosan Modification, %	CMC, μg/mL	Size, nm	Drug Loading by Mass, %	Time of Half-Release of Drug, h	Pharmacokinetics Parameters, Free Drug vs. Micellar Formulation
Present work	Chit5-OA-5 Chit5-OA-15 Chit5-OA-30 amCD-OA	5 15 30 14	0.18 0.03 0.03 0.03	112 100 89 132	40 44 48 35 MF	23 56 108 1.6 Free—0.65, PBS, cut-off 7 kDa	Cmax, μg/mL: 1.7 vs. 3.8 AUC_0–6h_: 8 vs. 16 MF
Y.-Z. Du et al. [27]	Chit8-LA-3.3 ** Chit8-LA-5.1 Chit8-LA-6.1 Chit14-LA-5.5 Chit20-LA-4.6	3.3 5.1 6.1 5.5 4.6	50 5 1 10 10	197 162 151 202 214	13 14.5 15 15 15 Doxorubicin	9 10 14 8 8 PBS, cut-off 3.5 kDa	-
R. Kumar et al. [36]	CM-Chit-OA ***	-	1	140	20.2	8/15 (pH 6.8/1.2), cut-off 14 kDa	C_max_, μg/mL: 0.07 vs. 0.14 AUC_0–72h_: 1.8 vs. 4.8 docetaxel
Barros et al. [38]	PLGA-Dex10 ****	-	620	113	-	-	-

* Designations from literary sources are adapted to the ones used by the authors: ChitX-OA-Y, where X = 5 kDa, molecular weight of chitosan, Y–OA: (monomers of chitosan) molar ratio in percent, in other words, modification degree; ** LA—linoleic acid; *** CM—Carboxymethyl; **** PLGA—poly(d, l-lactide-co-glycolide), Dex10—dextran 10 kDa.

## Data Availability

The data presented in this study are available in the main text and Appendix A.

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
