# Peer review of "Chitosan or Cyclodextrin Grafted with Oleic Acid Self-Assemble into Stabilized Polymeric Micelles with Potential of Drug Carriers"

_life, 2023, doi:10.3390/life13020446_

Round 1
Reviewer 2 Report
Dear Editor
I have reviewed the manuscript entitled "Polymeric micelles based on chitosan or cyclodextrin grafted 2 with oleic acid to enhance the efficiency of antibacterial drug". This article presents relevant information about the production and characterization of novel polymeric micelles using common polymers for drug delivery by covalent modification of chitosan and cyclodextrin with oleic acid. The manuscript is attractive, with promising findings, well presented, and adequately discussed. I suggest some changes before being accepted for publication:
1. Line 32. Which more advantages is been shown?
2. Line 52. I suggest changing ‘medicine’ for drug.
3. Line 54-57. Rewrite paragraph is confusing.
4. Line 65. Destruction? is this the best word?
5. Line 78. Reference?
6. Line 94. Are authors creating a new methodology to analyze micelles?, please clarify.
7. Purpose of sonication on micelle synthesis.
8. Line 173. Is this part of the results?
9. Line 174. What the authors means with micelle ‘destruction’.
10. Line 177. What does the author mean by sigmoid inflection? Where can I see those shifts?
11. Line 187. Author means ‘starved’ 12 h before the experiment?
12. For FTIR results, authors should label all the significant peaks in the plot, which is challenging for the reader trying to guess, especially amide bond formation.
13. Please add a reference for the methodology followed in lines 237-239 and explain how this is analized. It is challenging to observe the carbonyl group shift in order to claim micellar system formation.
14. Line 250-251. What is for MM and NTA, please clarify.
15. How did the authors determine the molecular weight of one structure unit polymer? I suggest running NMR analysis to confirm. How can authors ensure the %modification of chitosan with OA?
16. Please explain in more detail Figure3c. ‘The intersection point roughly corresponds to the CMC’.
17. Please try to discuss micelle formation with OA. Why do greater % polymer modifications show lower CMC and higher loading capacity?
18. Can loading capacity analyzed by FTIR be correlated with, for example, HPLC results?, is it not more practical to use another analytical technique?
19. What do the authors mean by polymer excess in Table 2?
20. Line 401. Please clarify how chitosan or CDextrin affects the drug release phase transition.
21. Figure 7. Please clarify the mechanism for drug release from this novel polymer micelle system.
22. Suggest changing word ‘destroyed’ in line 464.
23. Line 478-480, eliminate.
24. How chitosan is showing this PK effect, and no is observed on B-CD
25. Suggest an Electronic microscopy technique or AFM for sizing and micelle analysis.
26. CD micelles results are barely discussed.
Reviewer 3 Report
Major Revision

Reviewer 4 Report
The manuscript is nicely presented and all the results are supported with experimental observation. It may be accepted for the publications with minor corrections. The title of of the manuscript entitled "Polymeric micelles based on chitosan or cyclodextrin grafted with oleic acid to enhance the efficiency of antibacterial drug" is little bit confusing. It may be revised. The authors use both chitosan and cyclodextrin for the studies.
Round 2
Reviewer 1 Report
The reviewer's questions have been addressed and corresponding changes have been made. Overall the quality of the manuscript has been improved and can be recommended for Life.
Reviewer 3 Report
Accepted from my side.